# Dynamics of Household Waste Segregation Behaviour in Urban Community in Ujjain, India: A Framework Analysis

**DOI:** 10.3390/ijerph19127321

**Published:** 2022-06-15

**Authors:** Krushna Chandra Sahoo, Rachna Soni, Madhanraj Kalyanasundaram, Surya Singh, Vivek Parashar, Ashish Pathak, Manju R. Purohit, Yogesh Sabde, Cecilia Stålsby Lundborg, Kristi Sidney Annerstedt, Salla Atkins, Kamran Rousta, Vishal Diwan

**Affiliations:** 1Health Technology Assessment in India (HTAIn) Regional Hub, ICMR—Regional Medical Research Centre, Bhubaneswar 751023, India; sahookrushna@yahoo.com; 2R D Gardi Medical College, Ujjain 456001, India; sonirachana.n@gmail.com (R.S.); vivantima@gmail.com (V.P.); drashish.jpathak@gmail.com (A.P.); manjuraj.purohit64@gmail.com (M.R.P.); 3ICMR—National Institute of Epidemiology, Chennai 600077, India; madhanraj.k@icmr.gov.in; 4Division of Environmental Monitoring and Exposure Assessment (Water and Soil), ICMR—National Institute for Research in Environmental Health, Bhopal 462030, India; suryasingh.nireh@icmr.gov.in; 5Department of Global Public Health, Health Systems and Policy (HSP): Improving Use of Medicines, Karolinska Institutet, 171 77 Stockholm, Sweden; cecilia.stalsby.lundborg@ki.se; 6Division of Environmental Epidemiology, ICMR—National Institute for Research in Environmental Health, Bhopal 462030, India; sabdeyogesh@gmail.com; 7Department of Global Public Health, Social Medicine Infectious Disease and Migration (SIM), Karolinska Institutet, 171 77 Stockholm, Sweden; kristi.sidney@ki.se (K.S.A.); salla.atkins@tuni.fi (S.A.); 8Health Sciences and New Social Research, Faculty of Social Sciences, Tampere University, 33100 Tampere, Finland; 9Swedish Centre for Resource Recovery, University of Borås, 505 32 Borås, Sweden; kamran.rousta@hb.se

**Keywords:** household waste segregation, municipal solid waste, community perception, motivation-opportunity-ability-behaviour

## Abstract

Waste segregation practices must be socially acceptable, affordable, context-specific, and participatory, which is essential for promoting waste segregation. Therefore, this study explored the urban community members’ motivation, opportunity, and household waste segregation ability. We performed a qualitative study in Ujjain city, India. Ten focus group discussions and eight in-depth interviews were conducted with female and male household members in residential and slum areas. All interviews were digitally recorded, transcribed, and translated. We used the thematic framework technique using the Motivation-Opportunity-Ability-Behaviour theory for analysis. Three themes were constructed: motivation, where household members are motivated to sort waste yet fear the consequences of improper sorting; ability, where household waste segregation is rapidly gaining acceptance as a social norm; and opportunities, involving convenient facilities and a social support system for household members towards waste segregation. This study contributes to developing a knowledge base on waste segregation behaviour and a repertoire to facilitate evidence-based management and policymaking. There is a need for educational intervention and women’s self-help groups’ involvement to develop community orientation and waste segregation literacy. Finally, this study emphasizes the importance of all three behavioural change components, i.e., motivation, opportunity, and ability, in managing sustainable waste segregation practices.

## 1. Introduction

Every year, over two billion tonnes of municipal solid waste are generated globally, with at least 33% of that not being managed in an environmentally safe manner. Waste generated per person per day globally averages 0.74 kg but varies greatly, ranging from 0.11 to 4.54 kg. Solid waste management in cities in low-and middle-income countries (LMICs) is challenging [1,2,3]. Improper disposal of household waste has a wide range of environmental and health consequences. As a result of these expanding ecological, social, and economic concerns, waste management is increasingly recognised as a pressing issue by governments, businesses, non-governmental organisations (NGOs), academics, and the general public. Despite a growing body of research, little is known about the underlying factors that promote, drive, or obstruct waste segregation behaviours and practices, particularly in the context of LMICs.

Solid waste management (SWM) is a major issue in India, where urbanisation, industrialization, and economic growth have increased waste generation per person. Effective SWM is difficult in densely populated areas. SWM is a significant concern for many Indian urban local bodies (ULBs) especially in densely populated cities. To address this concern, the Government of India launched the Swachh Bharat Mission in 2014, followed by its second edition for urban areas in 2021 [4]. The program is being implemented by the Ministry of Housing and Urban Affairs. Its objective is to ensure hygiene, waste management, and sanitation across the urban areas of the country. Robust evaluation followed by standardised certification is being granted to the cities by garbage free city certification and Swachh Survekshan under the Swachh Bharat Mission. Star rankings are granted to cities with Garbage free city certification. This is a holistic evaluation of the cities across the entire SWM chain, where parameters include door to door waste collection at ward level, source segregation, preparedness of cities in scientific disposal of all waste categories and its use, as well as innovations and citizen participation among others [5]. Garbage free certification is one of the parameters for Swachh Survekshan (annual cleanliness survey) under the Swachh Bharat Mission; the others being open defecation free status and wastewater treatment. Behaviour change through people’s movement is a critical component of the strategy under SBM-U [4]. The ULBs are encouraged to distribute colour-coded bins (two bins per household) to separate waste at the source. Green bins are recommended for wet waste (i.e., biodegradables), while blue bins are recommended for non-biodegradable and other types of waste. Sanitary waste is recommended to be wrapped in paper and disposed separately, and hazardous waste is recommended to be disposed of separately and processed scientifically [6].

The concept of sustainable development incorporates economic, environmental, and socio-behavioral research activities [7,8,9]. As a result, it is necessary to develop a link between waste segregation and the existing socio-behavioural context, which will reflect the environmental impact of individual lives. Appropriate waste segregation is an ecological habit that affects both the built and natural environment. Pro-environmental behaviour or environmentally sustainable behaviours include responsibly engaging in home waste segregation with an adaptive reaction, characterising behaviour as a specific conduct that is useful to the environment [7,8,9].

Sorting and segregating household waste is vital in implementing successful and sustainable waste management systems [10]. The first step in waste management is waste segregation. Waste must be classified as biodegradable or non-biodegradable in order to facilitate waste management. However, very few households separate their waste. Evidence-based interventions that are socially acceptable, cost-effective, context-specific, and participative are needed [11,12,13,14]. Thus, understanding current household waste management methods are crucial for encouraging or changing proper waste sorting and segregation [15,16].

Behavioural economics has provided rich theoretical insights into human behaviour, particularly those involving judgement and decision making, over the last decade or so. Understanding how to promote more sustainable behaviours (including, but not limited to, recycling) in a variety of contexts continues to be a major challenge for policymakers and researchers. We conducted preliminary qualitative research to explore the range of influences on waste segregation behaviours in order to inform a subsequent quantitative survey because waste reduction behaviours have received little attention outside of the domestic context. This sequential mixed-methods approach has the advantage of using quantitative measures that are contextually relevant. There is, however, scant data concerning community behaviour, attitudes, and practices regarding waste segregation. Therefore, in this study, we explored the community members’ behaviour and its maintenance regarding motivation, opportunity, and ability for household waste sorting and segregation, with appropriate methods, findings, and relevant existing information. 

## 2. Materials and Methods

### 2.1. Study Design and Settings

We performed a qualitative study to document household members’ understanding of how inhabitants think and reflect on their waste management system in their daily life. This study was conducted in Ujjain Municipal Corporation, Madhya Pradesh, India. Ujjain has a population of 560,000 people and is divided into six zones and 54 administrative wards with 103,000 households, with approximately one quarter of the households located in slums [17]. Every day, nearly 226 tons of municipal solid waste is generated in Ujjain [18]. Ujjain has received 3-star rating for the year 2020 (7-star rating being best performance) in the garbage free city ranking and 4th rank in cities (of population 1 lakh and above) in Swachh Survekshan in 2021. This shows the scope for improvement in waste management for Ujjain. Even though this study was conducted in a single city in India as a pilot, the study intends to scale up in other cities in India and other similar contexts and to design and advocate for behavioural change communication interventions for household waste segregation.

### 2.2. Study Participants

This study recruited 78 participants using maximum variation sampling. Participants were selected purposefully based on the following: residential areas, such as residential households and urban slums, sex, and age group.The above individuals were chosen because they were largely involved in household waste segregation in their daily lives.We approached participants face-to-face.Thirty-five were from slums and 43 from non-slum residential areas; 47 were female, and 31 were male. The mean age of the participants was 41 years. About 23% had no formal education, 23% have completed primary school, 40% high school, and 14% graduated and above.We approached 92 people and 14 refused to participate due to their busy schedule.

### 2.3. Data Collection Procedure

Ten focus group discussions (FGDs) were held. Six FGDs with non-slum residents, two male and four female groups, and four FGDs with slum residents, two male and two female groups, were facilitated. The FGDs and IDIs were conducted during the period January–June 2021. Between six and eight participants participated in each FGD. Additional, eight individual interviewees (IDIs), five females and three males, were conducted to complement the FDGs. We decided on the number of FGDs and IDIs based on data saturation, and we collected and debriefed the data simultaneously to move on to the next round of data collection. We used FGD guide for data collection (Table 1), the tool was translated into local language, in this case ‘Hindi’, and the tool was pre-tested. The field team was trained in familiarity with the tool and facilitation of FGDs/IDIs by the authors, KCS and VD.

The authors RS and VP facilitated FGDs along with other field investigators, as they were native to the study setting and conversant with the local language. The facilitators had prior field experience and training in waste segregation and qualitative interviewing techniques. Before starting the FGDs and IDIs, we established a rapport with the participants through a local community-based organization. All participants were informed about the researchers and the research topic during the FGDs/IDIs. The facilitators bracketed the preunderstanding to avoid personal interest. The interview guide (Table 1) was developed based on the Motivation-Opportunity-Ability-Behaviour (MOAB) framework [19,20,21], entailing eight open-ended questions with probes.

All FGDs took place in the wards’ community halls. There were no other people present during FGDs, except the participants and facilitators. We did not conduct any repeat interviews. Throughout the FGDs, an observer kept field notes and memos. FGDs were conducted until data saturation, and we found that no new information was emerging with new FGDs. Each FGD lasted between 45 and 70 min. The average time for IDIs was 25 min. 

### 2.4. Data Management and Analysis

We conducted FGDs and IDIs in the local language, Hindi, and recorded them. We transcribed the digital data and translated the Hindi transcripts into English. We analysed data using the framework technique [22]. Repeated readings of the transcripts were performed by the authors KCS, MD, VD, KSA, and SA. Following familiarisation, the researcher read the transcript line by line and coded it. We adopted the MOAB theory and framework from previous studies to conduct a deductive framework analysis [19,20,21]. The MOAB theory proposes behaviour change based on motivation, ability, and opportunities. Motivation is defined as an individual’s goal-directed arousal or desire to behave in specific ways. Motivating factors include perceived benefits and threats, outcome expectancies, and goal setting. Ability towards waste sorting and segregation refers to the household members’ skill on waste segregation, which is compounded by proficiencies, knowledge, self-efficacy, self-monitoring, and coping strategies. Opportunities are the external or environmental factors that help in behavioural change. These are the environmental mechanisms, accessibility to, and availability of various resources and social support. 

We used the MOAB analytical framework to chart data into the framework matrix (MAXQDA Analytics Pro 2020, VERBI GmbH, Berlin, Germany). Authors KCS and RS coded and summarised the data using the MOAB framework. Finally, all authors contributed to data interpretation and analysis. We adopted cross-checking procedures during transcript analysis, whereby vernacular (Hindi) and English transcripts were used in tandem and cross-checked during the coding process to garner the transcript’s in-depth context. The authors’ varied educational and professional backgrounds and their experience in public health research allowed for a more nuanced understanding of the findings. Ten participants were debriefed on the findings following preliminary analysis to ensure member validity. We discussed our key findings independently with five slum dwellers and five non-slum residents to ascertain whether they agreed with our interpretation of the findings. We reported the study using the Consolidated Criteria for Reporting Qualitative Research (COREQ) guideline [23]. 

### 2.5. Ethical Considerations 

Ethical approval was obtained from the Institutional Ethical Committee of National Institute for Research in Environmental Health, Bhopal (NIREH/BPL/IEC/2020-21/41 dated 21 April 2020) and the Institutional ethics committee of R D Gardi Medical College, Ujjain (03/2020 dated 12 March 2020). The participants were assigned a unique identification number and pseudonymized. The interviews were audio-recorded with the participants’ permission. Before the FGD, we received written consent from all the participants. The study participants’ participation was entirely voluntary, and they could withdraw from the interview at any time without explaining.

## 3. Results

Three themes were constructed (Figure 1) to describe the findings: (1) motivation, where household members are motivated to sort waste yet fear the consequences of improper sorting; (2) ability, where household waste segregation is rapidly gaining acceptance as a social norm; and (3) opportunities, involving convenient facilities and social support system for household members towards waste segregation.

### 3.1. Theme 1: Motivation—Household Members Motivated to Segregate Waste Yet Fear Consequences of Improper Sorting

#### 3.1.1. Category: Perceived Threats and Benefits

The participants understood the benefits of sorting and segregating household waste. Simultaneously, they were conscious of the consequences of inappropriate segregation. They were concerned that improper waste segregation and disposal would cause health consequences, including breeding mosquitoes and flies, resulting in spread of diseases such as fever, malaria, dengue fever, and typhoid. Several of the participants also cited the annoyance of flies and mosquito bites. They regarded the spread of illness as threat, which would affect their health and the health of their community. They were particularly concerned about the health of their children.

“Mosquitoes will come if there is dirt and garbage in the house, and wet waste attracts more mosquitoes. Mosquito bites can cause dengue fever among many other diseases.”(FGD, 42 years female, non-slum)

Cleanliness was recognised as one of the significant benefits of waste segregation and appropriate disposal by many participants. Participants also understood the dispersion of dangerous compounds as a result of dry waste burning. Their children who were mainly in upper primary or high school-influenced the participants by informing them about the negative repercussions of garbage burning. Some individuals were also aware of environmental contamination, such as air, soil, and water contamination. The participants remarked how people dump waste into water bodies and clog them up, thus impairing land fertility.

“The environment will have an impact. If we do not throw plastic here and there, the land will not become barren. Water will accumulate adequately. We would suffer a massive loss due to our mistakes.”(FGD, 28 years male, slum)

#### 3.1.2. Category: Outcome Expectancies and Goal Settings

Some respondents from slums demanded that the municipal corporation give them a reward or compensation in recognition of appropriate waste segregation. In contrast, respondents from non-slum areas saw it as their civic duty to segregate. A few respondents stated that it would be a source of pride to keep the city clean by engaging in cleanliness-related behaviour, like their neighbouring city.

At the same time, some suggested penalties to households who were not segregating their household waste properly and felt that societal pressure would be effective in changing their neighbours’ behaviour. Some even suggested installing CCTV (closed-circuit television) cameras to monitor everyone’s garbage-related activities. There were indications that segregation might be gradually becoming a social norm. The respondent from a non-slum area described it as bad manners to dump garbage wherever one wants. Many younger female participants observed and described that after waste segregation was implemented, a difference in household cleanliness and less street litter would motivate them to work toward goal settings of household waste segregation.

“We’re doing it for ourselves, not for the government; if we live well, the country will be a better place to live. Our children would have a good life.”(FGD, 26 years female, non-slum)

Many participants stated that the work of segregation and disposal is not difficult if they receive two dust bins. Household members were concerned about residents who did not adhere to the segregation process. Participants from non-slum settings proposed forming a local committee of 4–5 household members to monitor waste segregation practices for the sake of community benefits. Many slum dwellers mentioned that they only have one bucket for waste collection.They put their dry waste in a bag and their wet waste in the bucket.

“We have begun to segregate; everyone understands and is aware of it. Those who are aware are keeping two waste bins. We will educate them on proper waste segregation and understand the importance of segregating waste near the waste collection vehicle.”(FGD, 45 years male, slum)

### 3.2. Theme 2: Ability—Household Waste Segregation Is Rapidly Gaining Acceptance as a Social Norm

#### 3.2.1. Category: Knowledge, Skills, Proficiencies and Self-Efficacy

Most participants were aware of the two dustbins for dry and wet waste. They were also knowledgeable about waste classification and segregation. Everyone agreed on the importance of waste segregation. Some of the households created compost from the wet waste and used it in their kitchen garden. Some recycled dry waste, such as paper and plastic. Several of them made crafts from waste bottles for decorative purposes. They kept children’s diapers, sanitary pads, and hazardous wastes eparately for disposal.

“Segregation has several advantages. We sell dry waste to scrap buyers every month. We do not mix hazardous waste like electronics products as well as waste medicine in our regular waste bin; occasionally, medicine waste is generated. It is kept outside in my area; we do not allow children even to touch it. We sell it to waste buyers.”(FGD 26 years female, non-slum)

Many of the household members have developed the habit of segregating wet and dry waste. Bottles, cans, clothing, plastic, wood, glass, metals, and paper were considered dry waste. They sold dry waste items, such as paper, cans, bottles, and clothing, to scrap buyers. Additionally, they exchanged their old clothes with scrap cloth buyers to get utensils. The household members have created a separate bin for waste napkins, sanitary pads, diapers, and medicines. Many male household members believed that females were primarily responsible for household waste segregation.

“My daughter-in-law segregates the waste at home. In general, it is the women’s responsibility to segregate the waste. Women have been doing so since the beginning.”(IDI, 56 years male, slum)

“My wife segregates the waste in my home. I hardly practice.”(FGD 48 years male, non-slum)

Many participants were conscious about their earlier improper ways of disposing waste. However, they have become more aware about waste segregation since they have heard about the Swachh Bharat Mission. Children learned about proper waste management in schools and on television.

“My six years old girl has well understand about waste, she yells to inform when waste collection vehicle arrives.”(FGD, 34 years female, non-slum)

The participants informed that the ULB staff, community health workers, and community-based organisations, such as self-help groups, educated residents about household waste sorting. The female staff accompanied and educated household female members about waste segregation, as all household members generate waste. Additionally, participants stated that when community members bring mixed garbage, collection staff immediately segregate the garbage and instruct them on proper waste sorting. The municipality staff also provided their contact information for additional assistance.

“We educate everyone in our community because we want our ward to be the best in waste segregation. We sell whatever can be sold; whatever needs to be thrown, we separate and keep in our dustbins. It is necessary to separate the waste.”(FGD, 32 years female, non-slum)

#### 3.2.2. Category: Coping Strategies and Self-Monitoring

Many non-slum residents claimed that waste collection vehicles did not come on Sundays or holidays and that when their buckets were full. They resorted to improper waste disposal. Many slum residents expressed dissatisfaction with the collection vans as they were irregular. Unavailability of dustbins was cited as one of the reasons for improper disposal by some respondents. On the other hand, some households created dustbins using unused packets or spare buckets. Several of them demanded provision of free waste collection bins. During festivals and weddings, paper, plastic, disposable plates, and bowls made of dry and wet leaves were disposed off without segregation.

“During festival and marriage, bulk of garbage generates—people dump their plates and garbage. It is difficult to segregate the waste.”(FGD, 47 years male, non-slum)

“We have a bucket at home. When it becomes useless, we make it as dustbins.”(FGD, 28 years female, slum)

To make it easier for children to recognise the appropriate bins for sorting, some participants suggested providing colour coded containers like blue, red, and green. Some of the participants viewed that although there was the provision of distribution of colour-coded bins under SBM-U, they were not distributed to every household. Many participants stated that the supervisor who arrives with the waste collection van informs the community about the waste separation process. Furthermore, they want this to be done more consistently and effectively.

“Someone should educate us about the importance of segregation and the processes involved, particularly to women who handle the household waste. They should also disseminate information door to door at each household periodically. They should arrive between 12 and 1 p.m. when everyone is free. Several community level meetings are scheduled; notify everyone that a penalty will be imposed if they cannot attend.”(FGD, 48 years male, non-slum)

### 3.3. Theme 3: Convenient Facilities and Social Support Systems Create Better Opportunities for Household Members for Waste Segregation

#### 3.3.1. Category: Convenient Facilities—Environmental Mechanisms, Accessibility and Availability

There were several missed opportunities for promoting and changing waste sorting behaviour that can be capitalised on. Due to the lack of a garbage van or garbage collectors at their doorstep, some residents dispose off their waste improperly by throwing it out in the open, burning dry waste, or dumping wet waste in an open sewer. Such a situation arose when the van was unable to navigate the narrow streets, or was overloaded with huge garbage during occasional household gatherings and festivals. Many a times, they segregated it and placed it in a polythene bag before throwing it into the garbage van in the adjacent street; occasionally, they throw it out in the open

“As our house is on a narrow road, the vehicle does not come at the doorstep; when we reach the vehicle, it has already left.”(FGD, 34 years female, non-slum)

The respondents expressed concern about the demeanour of frontline personnel. They emphasised the importance of friendly workers who would be wise to assist individuals in improving segregation practices. When asked, household members found it disrespectful to segregate trash in front of their neighbours, and they were worried about being judged for their waste. They mentioned the need for adequate information on waste segregation and creative activities such as manure preparation and recycling. They requested a convenient time for providing information so that more individuals would have access to it. Even though some slum dwellers lacked dustbins, they began segregating waste in oil cans, buckets, and large bowls. They did, however, want separate dustbins. Accessibility was compensated by an equitable distribution of segregation and disposal resources across a diverse population.

Although all members of households opined that they received information on segregation via schools, television, frontline workers, and neighbours;they acknowledged that the information provided to them was inadequate. They demand more orientation, and awareness is essential.

“When one person does it, others start following.”(IDI, 29 years female, slum)

“We will segregate it. They must explain it to us. Not everyone knows how to separate the waste.”(FGD, 56 years male, non-slum)

Many households were aware of the service charges associated with waste disposal. Some respondents found purchasing required number of dustbins financially burdening, amounting to inaccessibility. In the instances where there were narrow streets, residents must either walk to the van or wait for a waste collector to collect bags from them. They had informal access to some recycling facilities, such as scrap dealers who purchased newspapers and other glass and metal materials. Old clothing was exchanged for utensils or distributed to the needy.

“If there are two dustbins kept, then even a child will throw the waste in them.”(FGD, 37 years male, slum)

Adequate waste segregation requires easy access to waste segregation and disposal resources at the household level. Numerous households made dustbins out of cans, containers, buckets, or pans. The small size of such bins leads to their overfilling through the day. This was the foundation for their expressed need for waste collection facilities twice a day. Some household members were provided with dustbins that lacked lids.

“Send the waste collection van twice so that the garbage bucket does not overfill, or else a person tends to mix the garbage.”(FGD, 39 years male, non-slum)

Disposal van had separate bins for wet and dry waste, sanitary waste and hazardous waste and an announcement system. It stopped at most people’s doorstep. Some respondents were unaware of the existence of separate container for sanitary waste in the disposal van. Several households suggested a separate collection vehicle for food waste, which could be fed to cattle.

“Vegetables and sustenance appropriate for cows are tossed into the garbage van. There should be separate van, so the street animals can eat.”(FGD, 35 years female, non-slum)

The majority of households expressed satisfaction with the assistance provided by the disposal van driver and accompanying workers. The collectors were adamant about not accepting mixed garbage and would instruct families on properly segregating it. Pots were distributed to some for making compost, and workers visited them with information about the process.

“Everyone throws it in the van. If someone cannot understand, then the van driver explains to them; he helps in segregation.”(FGD 37 years female, non-slum)

#### 3.3.2. Category: Social Support System

Social support comprises support from the family and from other stakeholders who can advocate for changes in segregation-related behaviour and practice. Children are being taught about waste segregation as part of their school curricula, and hence most families benefit from this. Only a few kids could not segregate garbage effectively. Residents of a non-slum neighbourhood were urged by the authorities to actively monitor the neighbourhood for proper waste segregation. Numerous respondents were aware of their neighbours using a similar system of waste segregation with two dustbins.

Some participants said that the households should be rewarded for improved segregation. A few women suggested getting a certificate of appreciation. Officers of the ULB conducted supervisory and monitoring visits throughout the communities. Other agencies and workers visible to residents included the NGO workers who arrive with the waste disposal van, the Anganwadi employees (community workers for nutrition, health and education for women and children) who discussed segregation and reuse, and school professionals who pay visits to their neighbourhood. Several members stressed the importance of involving women self-help group members in community orientation and waste segregation literacy. According to several participants, a limited number of households were aware of waste segregation before implementing SBM-U. They believe that cleanliness has improved, while community attitudes and self-efficacy toward waste segregation have gradually improved.

“We were not doing so previously. Even if they have information, many people do not do it knowingly. However, many households segregated waste after the Swachh Bharat Mission began.”(FGD, 45 years male, non-slum)

## 4. Discussion

The participants were aware of perceived risks and benefits associated with household waste segregation. The household members, particularly the female members, gained skills and proficiencies, e.g., the habit of segregating wet and dry waste, and engaged in sorting and segregation, despite the lack of precise goal setting and consistency in waste segregation practises. The household members sorted and sold a variety of dry waste to scrap dealers. The majority of families indicated satisfaction with the waste disposal collecting van driver and accompanying personnel. Nonetheless, it overlooked various opportunities, such as the provision of colour-coded bins, waste collection vehicle accessibility, and proper educational intervention on waste sorting behaviours. The van could not manoeuvre through the small streets, was overloaded with massive waste generated during occasional household gatherings then some residents improperly disposed off their waste by dumping it in the open or by burning waste. Although, ULB staff, community health workers, and community-based organizations informed households about waste sorting, residents demanded better instructions at suitable timing.

Additionally, our findings indicate that segregation is becoming a matter of convenience and habit, though the extent of segregation varies. In the current situation in the Indian context, females are viewed as the primary stakeholders. Thus, it is necessary to train and motivate other male household members and instil the belief that waste segregation is not solely a female responsibility but that all household members share an equal role and responsibility. Our findings indicate that involving men in household waste segregation can ensure that everyone reflects a similar degree of responsibility. Thus, a more gender-focused capacity-building training would be highly beneficial in the future. Subjective norm, followed by attitude, is the most powerful predictor of behaviour change. Understanding and accepting social norms, especially the behaviour of family members, can help to adopt recycling behaviour [24,25]. We train children on the segregation practices. Every member of the family generates garbage and should also act as a segregator. Moral principles can also motivate initiating segregation and recycling efforts, though their effectiveness diminishes as recycling becomes more accessible to citizens [26].

Motivation and capability towards waste segregation result in an intention to perform practices, which is the prelude to behaviour [27]. Knowing how to complete a task empowers individuals to feel more competent in executing that activity, which was also the case for our participants. Additionally, they reviewed the operational procedures for conveying such information as to timing, audience, and content. Goal setting and goal attainment is crucial to achieving the objective [27]. It has been established that environmental views have a varying effect on environmental behaviour [24]. However, a favourable attitude toward waste separation at the source can increase participation in recycling programmes [28]. As this study discovered, citizens’ sense of responsibility for their families and health also promotes pro-environmental behaviour [29]. The family’s health and well-being norms affect motivation, aligning with more substantial environmental and civic goals. Money, time, physical exertion, and space are all costs [27,30], but health, the environment, and financial security are all benefits [27,30]. Participants in our study evaluated waste segregation to have health benefits, including increased cleanliness and decreased contamination. Several responders stated that before SBM-U, just a few households knew about waste segregation. They feel the community’s attitudes and self-efficacy toward waste segregation have improved steadily. Repetition creates habits and encourages the community to follow [19].

Easy access to a segregated waste disposal system is crucial for better practices, proved in the various door-to-door waste pickup and recycling bin interventions [31,32,33]. Narrow lanes hamper doorstepping, a concern voiced by research participants. Various configurations in specific locales can ensure universal access. Separate colour-coded dustbins at the source (in households) were a perceived requirement for many, as many households collected waste in homemade dustbins of varied capacity that required frequent clearing. Additionally, colour coding can help guarantee that all family members, including children and senior residents understand segregation.

The trend in the generation of solid waste indicates an increase in the proportion of plastic waste [34]. Recycling recyclable materials such as plastics, metals, and glass have increased in recent years due to the increased consumption of packaged goods [35,36]. The enormous increase in plastic garbage and other waste results from changing lifestyles [37], which signifies the need for proper management of household solid waste. Reducing, reusing, and recycling waste can save money for customers [27]. Recycling and composting are enterprises that can provide additional financial or other incentives to citizens to segregate. Economic incentives may work better than social influence [38], and waste-for-goods exchange schemes may also succeed [16,39]. A few participants in this study wanted awards and acknowledgement for proper segregation. Our findings suggest that frontline staff behaviour, information and feedback from residents, and help from other social organizations, such as schools and anganwadis, provide necessary nudges. Nudging is based on the idea that individuals make decisions based on how things are presented to them, even if they have all the essential knowledge and have calculated the associated costs [40]. A nudge can be utilized to induce waste separation at the source [41].

Furthermore, several studies have found that nudging may be ineffective for people who sort more [42]. The way segregationists and non-segregationists see the enforcement of segregation laws differs [43]. Government employees are less adept at communicating with the public than community leaders [30]. There is little evidence of these leaders’ involvement in the current study.

In most LMICs, limited budget allocation for household waste management is the primary impediment to waste management [44], consistent with our findings. The initiatives incorporate elements of community development and the informal sector. Alternative waste management solutions consider both behavioural change communication and technical improvement. Behavioural change communication influences public behaviour by strengthening communities through training and fostering collaborations with decentralized waste management [45,46] with the assistance of non-governmental organizations (NGOs), local leaders, and facilitators working as trainers [47]. Technical solutions include reducing biodegradable waste at the source, converting waste to energy, and utilizing essential technologies. These strategies are expected to improve the sustainability of waste management.

The current research contributes to our understanding of the factors influencing community members’ waste segregation behaviour in our settings. It is intended to assist municipal governments and non-profit organizations in the sustainable management of household solid waste. Interventions with communication materials and messages designed based on current research findings and targeting households and schools can increase household waste segregation at source. Recognizing and appreciating citizens’ efforts may elevate their morale. Additionally, motivating household members through partnerships with recycling industries is critical for them to perform the task of segregation efficiently. This can assist individuals in making lifestyle choices that benefit the environment, health, and overall quality of life.

To enhance the trustworthiness of the study, we pursued research and source triangulation. Given our diverse academic, professional backgrounds and country, we interpreted the study results with broad perspectives. Participants came from different socio-economic backgrounds, with male and female participants from different settings and age groups. To minimize bias, the researchers’ preconceived notions were bracketed during data collection and analysis. While our study was conducted in the Indian context, the findings may apply to other similar LMICs settings.

## 5. Conclusions

Household members, especially women, obtained skills and proficiencies, e.g., the habit of separating wet and dry waste, and engaged in sorting and segregation practices. Our findings are significant for comprehending context-specific issues and advocating for resource-constrained settings to implement behavioural interventions on waste segregation. This study contributes to developing a knowledge base on waste segregation behaviour and a repertoire to facilitate evidence-based management and policymaking for the community members for effective waste segregation. The findings will help municipal employees and authorities build better strategies for household waste segregation practices. There is a need for educational intervention and women’s self-help groups’ involvement in developing community orientation and waste segregation literacy. Finally, this study emphasises the importance of all three behaviour change components, namely motivation, opportunity, and ability, in managing sustainable waste segregation practices.

## Figures and Tables

**Figure 1 ijerph-19-07321-f001:**
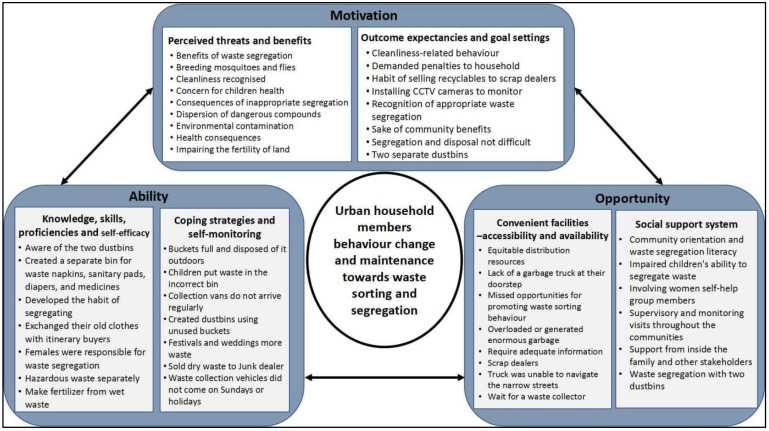
MOAB framework for household waste segregation behaviour and maintenance.

**Table 1 ijerph-19-07321-t001:** Focus Group Discussion guide.

**A.** **Existing Practice on Household Waste Sorting and Segregation** Can you describe your household waste? Probes: Types of waste generated and classification with examples including medicine, plastics etc., difference between dry and wet waste, and waste generated on daily, weekly, during festivals. 2. What are your current household waste management practices? Probes: Sorting and segregation, dumping, selling, recycling, and reuse, key responsible person in managing the waste, in absence/sickness of the key person, role of male versus female, children, and elderly in waste sorting, and storage process, containers, and if any seasonal variation
**B.** **Opportunity (Accessibility, Availability and Social Support)** Can you share your experience on existing support system for household waste segregation in your community/ward/city? Probes: Availability and accessibility of facility from municipality, incentive or household tax, awareness strategies, engagement of community-based organization, and informal sector 2. What is your expectation from other stakeholders for segregation of household waste? Probes: Information they need, effective information sources, and guidance, support from municipality, community-based organization or community leader, and informal sector
**C.** **Ability (Skill, Technology, Efficacy, Coping)** What are the challenges for sorting and segregation of household waste? Probes: Segregation skill, coping strategies, use of technology, logistics, time management, and skill/proficiencies, self-efficacy, self-monitoring 2. What types of support do you expect for sorting and segregation of household waste? Probes: Orientation and training, education, and incentives etc.
**D.** **Motivation (Outcome expectancies, perceived benefits/threats, goal settings)** Why proper disposal of waste is necessary? Probes: Environmental impact, physical health, and psychosocial well-being 2. What do you suggest for effective implementation segregation of household waste? Probes: Steps we should take to sustain segregation, community members‘motivation, support do you expect from municipality, and help from informal sector (selling, exchange, free, pay)

## Data Availability

The data will be provided as per request to the corresponding author with prior approval.

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
