# Peer review of "Dynamics of Household Waste Segregation Behaviour in Urban Community in Ujjain, India: A Framework Analysis"

_ijerph, 2022, doi:10.3390/ijerph19127321_

Round 1

Reviewer 1 Report

I suggest that the objectives be clarified in the beginning. 

Explain the reasons for selecting participants for the sample. 

Does the sample reflect the profile of the population?

Indicate who will use the results of the study

Author Response

Dear, Ms. Alex He, Editor

We thank the reviewers for their input. The reviewer's feedbacks are sincerely appreciated. Our revised manuscript is enclosed. We revised the manuscript accordingly and include the detailed response. We respond to the individual comments in the following: comments by the reviewers are indicated as RC and authors responses as AR. The modification/changes in the revised manuscript are also highlighted

Reviewer 1

RC: I suggest that the objectives be clarified in the beginning.

AR: As per the suggestion, we clarified the objective in Abstract (in lines 31-32) and in Introduction section (lines 111-114)

RC: Explain the reasons for selecting participants for the sample. 

AR: We explained in lines 133 & 134 in Methods section.

RC: Does the sample reflect the profile of the population?

AR: The samples reflect the profile of the population (lines 135-139).

RC: Indicate who will use the results of the study.

AR: We have revised accordingly in lines 530-540 in Conclusion section.

Author Response

Dear,   

Ms. Alex He, Editor

We thank the reviewers for their input. The reviewer's feedbacks are sincerely appreciated. Our revised manuscript is enclosed. We revised the manuscript accordingly and include the detailed response. We respond to the individual comments in the following: comments by the reviewers are indicated as RC and authors responses as AR. The modification/changes in the revised manuscript are also highlighted. The English editing of MS is also done by native speaker. 

Reviewer 2

RC: Waste segregation and its further processing are some of the main challenges of sustainable development that are related to ecology. The manuscript submitted for review addresses an exciting and timely topic on waste segregation. The structure of the manuscript and the literature selection are correct. However, I have some comments for the authors.

AR: Thanks for the suggestions.

RC: The Introduction section does not provide a proper background for the rest of the manuscript. The link between waste segregation and the current context of ecology is missing. It should not be forgotten that adequate waste segregation is undoubtedly an ecological behavior that affects our environment. However, it is not the only environmental behavior. At the beginning of the "Introduction" section, I propose to complement it with a short review of human pro-ecological behavior based on the latest literature: https://doi.org/10.3390/en15051690 ; https://doi.org/10.1177/0013916517701796 ; https://doi.org/10.3390/en14061767 

AR: We have revised accordingly and added the following reference [7-9] as suggested (lines 86-93).

RC: At the end of the Introduction section, please provide the manuscript's structure (individual sections). This will increase the clarity and readability of the manuscript.

AR: We have added in lines 111- 115.

RC: In the "Data collection procedure" section, there is no information on the period during which interviews were conducted and data collected.

AR: The period of data collection mentioned in lines 143 & 144, in Methods section.

RC: At the end of the "Conclusions" section, please indicate the limitations of the manuscript and opportunities for future research in this area.

AR: We have revised accordingly in lines 522-530 in Conclusion section.

Reviewer 3 Report

Dear Authors,

Some comments for you.

Line 119

Why only in India? 

Line 122

Why 78 participants? Which statistical method was used to determine this number 78?

Do the participants receive previous information to answer the questions in Table 1?

Only 8 questions for the discussion group?

It is not clear how the data were processed.

In the methodology section, the authors mentioned:

a)     Existing practices on household waste sorting and segregation

b)    Opportunity ….

c)     Ability….

d)    Motivation…

However, in the results section, they do not include letter a)

In addition, the authors said: “the participants understood the benefits of sorting…” How do you know that? How can you assure that they understood? 

      Why do the authors explain about composting in line 460? Was the subject part of your interview?

You have several variables; it is necessary to have a statistical analysis for each of them or at least some graphs, percentages, SD, etc., to show the behavior of each variable.

Why in the conclusions section there is no information about the existing practices?

Author Response

Dear,   

Ms. Alex He, Editor

We thank the reviewers for their input. The reviewer's feedbacks are sincerely appreciated. Our revised manuscript is enclosed. We revised the manuscript accordingly and include the detailed response. We respond to the individual comments in the following: comments by the reviewers are indicated as RC and authors responses as AR. The modification/changes in the revised manuscript are also highlighted.  English editing of MS is also done by native speaker.

Reviewer 3

RC: Line 119 Why only in India? 

AR:  We have modified accordingly in line 127.

RC Line 122 Why 78 participants? Which statistical method was used to determine this number 78?

AR: We used purposive sampling with maximum variation (non-probability) approach, which is best suggesting sampling methods in qualitative research.  

RC: Do the participants receive previous information to answer the questions in Table 1?

AR: We have added in lines 156 and 157.

RC: Only 8 questions for the discussion group?

AR: We have explained it in Lines 159-161.

RC: It is not clear how the data were processed.

AR: We have explained it in lines 185 – 190.

RC: In the methodology section, the authors mentioned: a) Existing practices on household waste sorting and segregation, b) Opportunity, c) Ability, and d) Motivation. However, in the results section, they do not include letter a)

AR: During the analysis the existing practices were merged in ability sections (lines 270-278)

RC: In addition, the authors said: “the participants understood the benefits of sorting…” How do you know that? How can you assure that they understood?

AR:  We explained it in lines 218-226.

RC: Why do the authors explain about composting in line 460? Was the subject part of your interview?

AR: We have removed the composting section as suggested.

RC: You have several variables; it is necessary to have a statistical analysis for each of them or at least some graphs, percentages, SD, etc., to show the behavior of each variable.

AR: Thank you for the suggestions. However, this may not be applicable here as we followed qualitative approach to answer our research question.  We employed open-ended questions to obtain textual data and then applied thematic analysis. However, in our next project follow-up study, while conducting quantitative study, we will describe  the magnitude of the problems using relevant descriptive and inferential statistical measures

RC: Why in the conclusions section there is no information about the existing practices?

AR: We have revised accordingly in lines 530-540 in Conclusion section.

Round 2

Reviewer 3 Report

I have no more suggestions. It is an interesting paper.